# Patterns of genomic evolution in advanced melanoma

E. Birkeland[1,2], S. Zhang[1,2], D. Poduval[1,2], J. Geisler[3,4], S. Nakken [5,6], D. Vodak[5,6], L.A. Meza-Zepeda[5,6,7], E. Hovig [5,6,8,9], O. Myklebost [5,6], S. Knappskog[1,2] & P.E. Lønning [1,2]

Genomic alterations occurring during melanoma progression and the resulting genomic heterogeneity between metastatic deposits remain incompletely understood. Analyzing 86 metastatic melanoma deposits from 53 patients with whole-exome sequencing (WES), we show a low branch to trunk mutation ratio and little intermetastatic heterogeneity, with driver mutations almost completely shared between lesions. Branch mutations consistent with UV damage indicate that metastases may arise from different subclones in the primary tumor. Selective gain of mutated *BRAF* alleles occurs as an early event, contrasting whole-genome duplication (WGD) occurring as a late truncal event in about 40% of cases. One patient revealed elevated mutational diversity, probably related to previous chemotherapy and DNA repair defects. In another patient having received radiotherapy toward a lymph node metastasis, we detected a radiotherapy-related mutational signature in two subsequent distant relapses, consistent with secondary metastatic seeding. Our findings add to the understanding of genomic evolution in metastatic melanomas.

[1] Section of Oncology, Department of Clinical Science, University of Bergen, 5020 Bergen, Norway. [2] Department of Oncology, Haukeland University Hospital, 5021 Bergen, Norway. [3] Institute of Clinical Medicine, University of Oslo, Campus Akershus University Hospital, 1478 Lørenskog, Oslo, Norway. [4] Department of Oncology, Akershus University Hospital, 1478 Lørenskog, Norway. [5] Department of Tumor Biology, Institute of Cancer Research, The Norwegian Radium Hospital, Oslo University Hospital, 0310 Oslo, Norway. [6] Norwegian Cancer Genomics Consortium, Institute for Cancer Research, Oslo University Hospital –Radium Hospital, 0310 Oslo, Norway. [7] Genomics Core Facility, Department of Core Facilities, Institute of Cancer Research, the Norwegian Radium Hospital, 0310 Oslo, Norway. [8] Department of Informatics, University of Oslo, 0316 Oslo, Norway. [9] Institute of Cancer Genetics and Informatics, The Norwegian Radium Hospital, Oslo University Hospital, 0310 Oslo, Norway. Correspondence and requests for materials should be addressed to P.E.Løn. (email: per.lonning@helse-bergen.no)

The incidence of melanoma is rapidly increasing among light-skinned people[1], where both epidemiological[2] and genomic evidence have established the link between melanoma etiology and UV radiation[3–5]. Many melanomas reveal an indolent course characterized by locoregional relapses followed by a rapid emergence of metastatic disease, and there is evidence suggesting that systemic dissemination may bypass intermediary stages of lymph node involvement[6,7].

Somatic mutations found in a cancer mirror its initiation and evolution, and genomic sequencing may thus map the progression of melanomas from earlier stages of development, enabling inferences that are empowered by comparisons of multiple lesions. While a few studies have used comparative lesion sequencing to assess genomic events during the process from benign lesions to primary melanomas[8] and progression from primary to regional disease[9], most studies of metastatic melanoma have explored genome evolution in response to targeted therapy[10–12]. A picture is emerging where most UV-associated mutations arise in the primary tumor prior to malignant transformation, followed by an increased frequency of copy number alterations[8]. The genomic events driving tumor progression toward advanced disease, however, remain incompletely understood.

Melanomas have low sensitivity to chemotherapy[13]. While recent developments including immune checkpoint inhibitors and BRAF/MEK targeting agents have improved the outcomes significantly, many patients do not achieve durable remissions[14,15]. Thus, improvements in therapy are needed. This may be facilitated by an improved understanding of genomic events associated with accelerated growth and dissemination.

Here we performed whole-exome sequencing (WES) of single or multiple metastases from a cohort of patients with advanced melanoma. Our findings add novel data to the understanding of the chronological sequence of genomic alterations. This includes early copy number gain of the mutated *BRAF* allele and the finding that whole-genome duplication (WGD) in general occurs as a late truncal event. While we found evidence indicating polyclonal seeding in one patient, this seems to be a rare event. Among four patients exposed to dacarbazine, we observed a "mutational signature" in one, probably related to several *MSH6* mutations in her tumor. Moreover, the finding that radiotherapy toward a lymph node metastasis may influence mutation signatures in subsequent deposits in organs distant from the treatment site supports the hypothesis that cancers may progress also through secondary spread from metastatic deposits.

## Results

**Single-base substitutions and indels**. We analyzed 114 metastatic lesions with matched normal tissue from 60 patients diagnosed with advanced melanoma by WES. All patients were from a prospective study assessing dacarbazine therapy for metastatic melanoma[16,17]. Eighty-six lesions from 53 patients consisting of at least 20% tumor cells (threshold for copy number profiling) were selected for further analysis (identified mutations in these samples are presented in Supplementary Table 1). Multiple lesions were available from 23 out of the 53 patients, and single-metastatic lesions were available from the remaining 30 (Table 1, and Supplementary Tables 2 and 3).

The number of somatic variants identified in coding regions per patient (average across samples for patients with multiple biopsies) varied substantially, with between 17 and 4089 mutations identified (range: 0.34–81.8 mutations per megabase, median: 9.6; Fig. 1a). With few exceptions, tumors with primary origins at sun-exposed sites all displayed mutational patterns characterized by C>T transitions at dipyrimidine sites, in contrast

| Table 1 Patient characteristics | |
|---|---|
| **Baseline characteristics** | **Patients** |
| Sex | |
| Female | 22 |
| Male | 31 |
| Disease origin | |
| Cutaneous (non-glabrous skin) | |
| Head | 5 |
| Upper extremities | 5 |
| Trunk | 20 |
| Lower extremities | 7 |
| Acral | 3 |
| Uveal | 2 |
| Mucosal | 2 |
| Primary unknown | 9 |
| Number of samples | |
| 1 | 30 |
| 2 | 16 |
| ≥3 | 7 |
| **Molecular characteristics** | |
| Mutational subtype | |
| BRAF | 27 |
| NRAS | 17 |
| NF1 | 2 |
| Triple wild type | 7 |
| Genome duplication | |
| Near-diploid | 32 |
| Genome duplicated | 21 |
| Total | 53 |

to tumors derived from areas not exposed to UV radiation (Fig. 1b), consistent with UV-induced DNA damage (Fig. 1c). One acral melanoma had a UV-associated mutational signature, as has also been observed by others[18,19]. Overall though, patients with sun-exposed primary tumors had a higher mutational load than patients with primary lesions at sites with little or no such exposure ($p < 0.001$, Mann–Whitney $U$-test [MW]; Supplementary Figure 1a). No difference in mutation load between the lymph node and subcutaneous or visceral organ metastases was recorded (Supplementary Figure 1b).

Among nine patients diagnosed with metastatic melanoma without known primary lesions, the types and numbers of mutations resembled those observed in metastases from sun-exposed primary lesions, strongly suggesting cutaneous origins (Fig. 1, Supplementary Figure 1a) as previously reported by others[20].

**Driver mutations and genomic complexity**. Using a conservative approach to identify driver mutations, we considered mutations in a set of predefined genes based on recently published studies[3,21,22]. Mutations in these genes were manually assessed to determine their status as drivers or passengers (Methods section). The complete list of mutations in these genes is presented in Supplementary Table 4. Driver mutations in *BRAF* and *NRAS* were detected among 28 (53%) and 17 (32%) patients, respectively (Fig. 1d), with one patient carrying a non-canonical driver mutation in *BRAF* (p.E586K) in combination with a driver mutation in *NRAS* (p.Q61L). While protein-altering mutations in *NF1* were identified in five patients, only two of these fulfilled our criteria for definition as driver mutation. Driver mutations in *GNAQ* and *GNA11* were identified in two uveal melanomas, and a driver mutation in *KIT* was found in mucosal melanoma.

Considering patients with multiple sampled lesions, all driver mutations identified were shared between metastatic deposits,

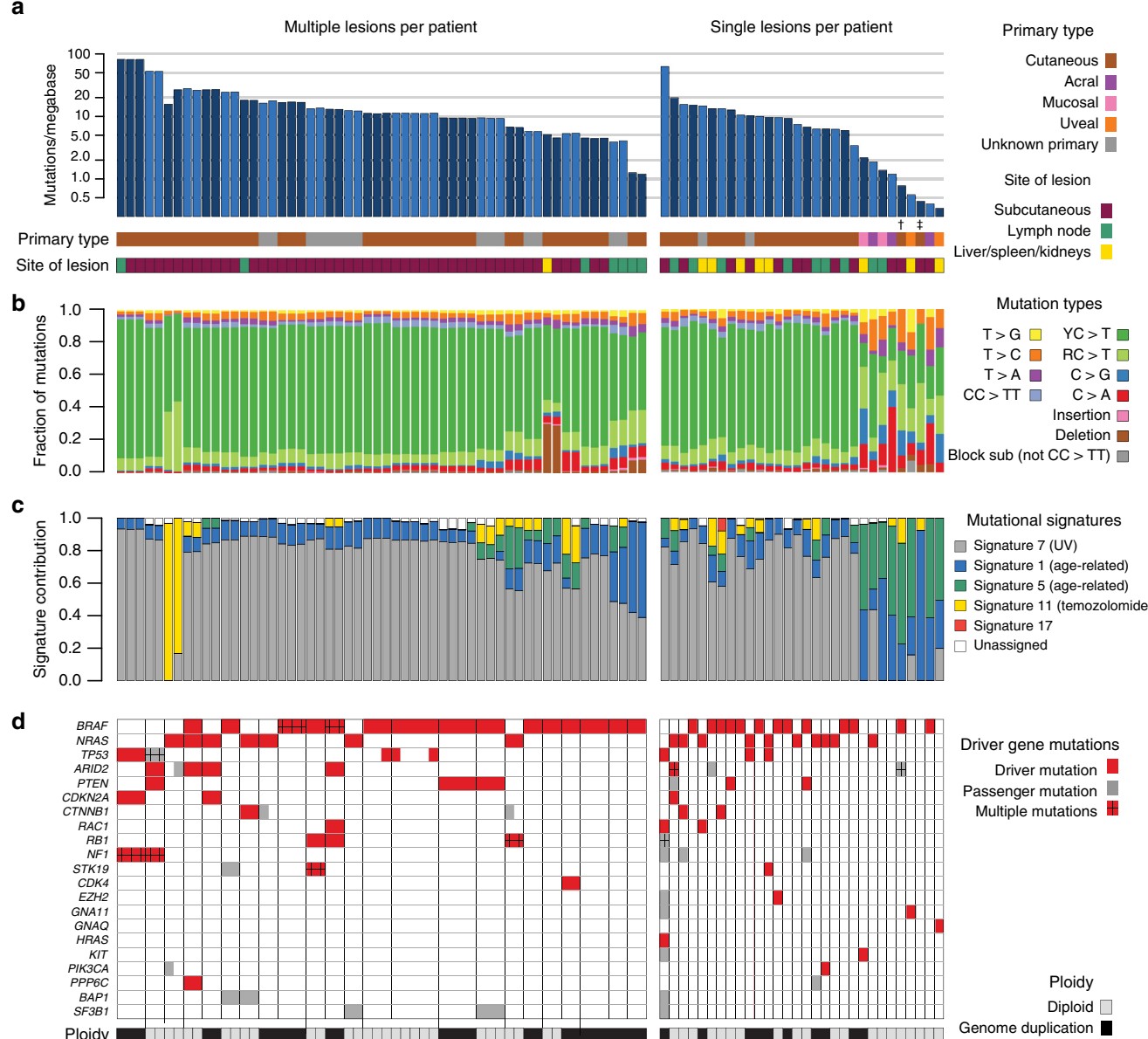

**Fig. 1** Overview of mutations. **a–d** Left: patients from whom multiple lesions were analyzed. Right: patients from whom single lesions were sampled. Patients are ordered by the number of mutations identified per patient, and lesions are further ordered according to time of sampling. **a** Number of mutations per megabase in each individual sample. Samples from different patients are indicated by alternating shades of blue. † One patient had a borderline acral primary tumor situated at a toe. ‡ One patient had a perianal cutaneous primary tumor, likely not exposed to UV radiation. **b** Fraction of mutation types per lesion. **c** Estimated contribution of mutational processes by fraction of mutations explained by each mutational signature[26], according to the classification of Alexandrov et al.[32]. Only signatures explaining >5% of mutations are shown, and only signatures 7, 1, 5, 11, and 17 were assessed. **d** Mutations identified per lesion in established melanoma driver genes are color-coded: red boxes indicate driver mutations and gray boxes indicate passenger mutations. Multiple mutations per gene are indicated with "+". Genome duplication events are shown per sample in gray (diploid) or black (genome duplication) for each sample

except for two patients, both revealing heterogeneous and subclonal distribution of the p.Y163C TP53 mutation.

In accordance with previous reports[22,23], we found the number of mutations to vary according to driver mutation status in *BRAF*, *NRAS*, and *NF1* ($p = 0.002$, Kruskal–Wallis rank-sum test [KW]; Supplementary Figure 1c). Based on copy number profiling (Supplementary Figure 2a), we inferred whole-genome duplication (WGD) events to have occurred in about 40% of patients (Supplementary Figure 2b), with no difference between tumors harboring *BRAF* (11/27) or *NRAS* (7/17) mutations. The duplication events likely predated

evolutionary divergence of metastases, as they were identified across all lesions obtained from these patients. Notably, the genomic complexity (defined as the fraction of the genome in an aberrant state, i.e., deviation from a balanced copy number of two for diploid tumors and four for WGD) was substantially higher in samples with WGD, with a mean of 69% for WGD and 30% for diploid tumors ($p < 0.001$, MW test; Supplementary Figure 2c). A difference in genomic complexity of this magnitude indicates a greater propensity for genomic alterations following genome duplication, as previously reported in other cancer forms[24,25].

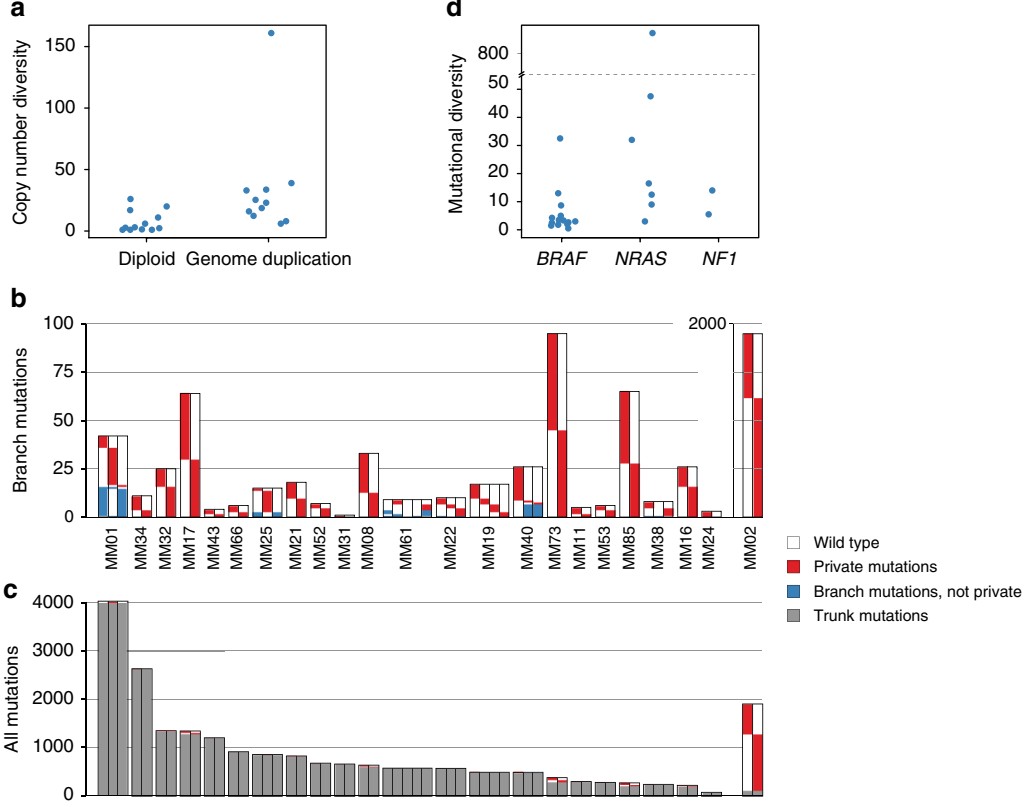

**Fig. 2** Mutational heterogeneity. **a** Copy number diversity according to whether genome duplication was identified in samples from each patient. **b** Branch mutations (found in more than one, but not all samples) and private mutations (found exclusively in one sample) per lesion. For MM02, a separate axis is used to capture the large number of private mutations. **c** All mutations (private, branch, and trunk mutations) presented together for each lesion. **d** Mutational diversity (average number of branch mutations per sample) in patients according to driver mutation status of *BRAF* and *NRAS*. The Y-axis is broken for clarity due to the high mutational diversity in MM02

**Heterogeneity across intraindividual lesions**. We also observed larger copy number diversity (defined as the mean number of copy number alterations separating samples from individual patients) in patients if WGD was present compared to patients with diploid tumors, where the median copy number diversity of patients with diploid and WGD cancers was 2.8 (range 1–26) and 23 (range 6–161), respectively ($p = 0.004$, Fig. 2a). This suggests the copy number evolution to be an ongoing process occurring at a higher rate in melanomas with WGD. Diversity in copy numbers was observed across the genome, with no chromosome being overrepresented ($p = 0.3$, KW test; Supplementary Figure 3).

In order to investigate the mutational heterogeneity between melanoma metastases, we identified trunk and branch mutations for each of the 23 patients having multiple lesions examined. Mutations were classified as trunk mutations when found in all lesions examined from a particular patient, or when the absence of a mutation could be explained by a copy number loss or lack of sequencing depth in a sample without this mutation. Branch mutations were accordingly defined as those mutations whose absence could not be explained by the same features. Branch mutations were further defined as private when exclusively identified in a single sample. Thus, we defined mutational diversity for each patient as the average number of branch mutations across lesions.

Patients generally displayed a low degree of mutational diversity (range: 0.5–893, median: 5) when compared to the number of trunk mutations (range: 17–3966, median: 465; Fig. 2b, c). Thus, with the exception of a single patient (MM02) whose metastatic deposits contained 89% branch mutations (probably related to chemotherapy exposure; see below), the branch

mutations for each individual patient accounted for only 0.08–14.9% of the mutation load. Notably, across patients, no correlation between the number of trunk mutations and mutational diversity was observed ($r_s = 0.01$, $p = 0.95$, Spearman's rank correlation).

While the number of mutations private to any lesion varied substantially (range 0–1156), the number of private mutations revealed a remarkable within-patient consistency, indicating an intrinsic propensity for mutational accumulation (Supplementary Figure 4). Excluding patient MM02, who had an extremely high number of private mutations in both lesions sampled, from statistical comparison, we found the degree of intraindividual variation across the sample set to be significantly lower as compared to interindividual variation ($p = 0.003$, Levene's test for homogeneity between groups). Assessing within-patient differences in types of branch mutations, we found small variations in mutation types related to private mutations across samples, as well as branch mutation types according to clonal status (Supplementary Figure. 5a and b), supporting mutational accumulation to be related to tumor intrinsic phenotypes.

Mutational diversity was significantly lower in tumors harboring a *BRAF* versus an *NRAS* or *NF1* mutation ($p = 0.01$, KW test; Fig. 2d). While this mirrored the difference in mutational load in general, the lack of correspondence between the number of trunk mutations and mutational diversity between tumors suggests these observations to be independent. No correlation between mutational and copy number diversity across patients was observed ($r_s = -0.07$, $p = 0.8$, Spearman), and copy number diversity was unrelated to *BRAF*, *NRAS*, or *NF1* mutational status ($p = 0.8$, KW test).

Categorizing patients into four groups based on the largest anatomical distance between sampled lesions (same site; different site, but same region; different regions; or different organ system), we observed no difference in either mutational or copy number diversity related to anatomic distance between the samples ($p = 0.3$ and $p = 0.7$, KW test; Supplementary Figure 6). Also, there was no difference in diversity between synchronous metastases and those collected with an intervening time period ($p = 0.5$ and 0.7, KW test, for mutational and copy number diversity, respectively).

**Shift in mutational processes**. Comparing trunk to branch mutations, there was a clear shift in the types of mutations between the two groups, with branch mutations being drawn from a much more widely distributed repertoire of mutation types (Fig. 3). All of the patients with multiple sampled lesions had primary lesions in sun-exposed locations (or unknown primaries) and, consistent with a history of sun-exposure, mutational signature analysis[26] revealed 42–93% (median 84%) of trunk mutations to belong to the UV signature (Supplementary Figure 7a). The limited number of branch mutations made any signature derivation uncertain. However, we observed a mutation pattern consistent with an UV signature in a total of six out of 14 patients (Fig. 3 and Supplementary Figure 7b), and in one patient (MM01), UV-related mutations was the major mutation type in the branches. In contrast, we observed no enrichment of UVA-associated T>G transversions[27,28].

**Evaluation of polyclonal seeding**. Studies of metastatic cancers including melanoma have unveiled polyclonal seeding and complex patterns of metastatic dissemination[9,29]. Applying the pigeonhole principle[30], the cellular prevalence of mutations can be used to infer the order of mutational accumulation and selective sweeps in populations of cancer cells. When comparing the cellular prevalence of mutations in two different samples of common ancestry, subclonal mutations shared across lesions may indicate polyclonal seeding, while the presence of lesion-private and clonal (defined as a mutation occurring in all tumor cells in that lesion and not in others) mutations would preclude such an interpretation and likely indicate a monoclonal origin.

We compared the relative variant allele frequency (rVAF; reflecting cellular prevalence) of private mutations in each lesion to that of trunk mutations (Fig. 4a, Supplementary Figure. 8). The rVAF distribution of trunk mutations was used to infer the likely clonal status of private mutations in each sample. Although many private mutations were clearly subclonal (e.g., MM17; Fig. 4a), 41 out of 53 samples revealed at least one clonal private mutation (Fig. 4b), implying an absence of polyclonal seeding. Only two patients (MM24 and MM31) lacked clonal private mutations altogether. Except for two mutations in MM31 having low rVAFs in both sampled lesions, these patients did not have shared subclonal mutations. Thus, we concluded that there was no strong evidence supporting polyclonal seeding in these patients either. Cross-sample mutation clustering, applying PyClone[31], corroborated these observations (Supplementary Figure 9). Yet, in

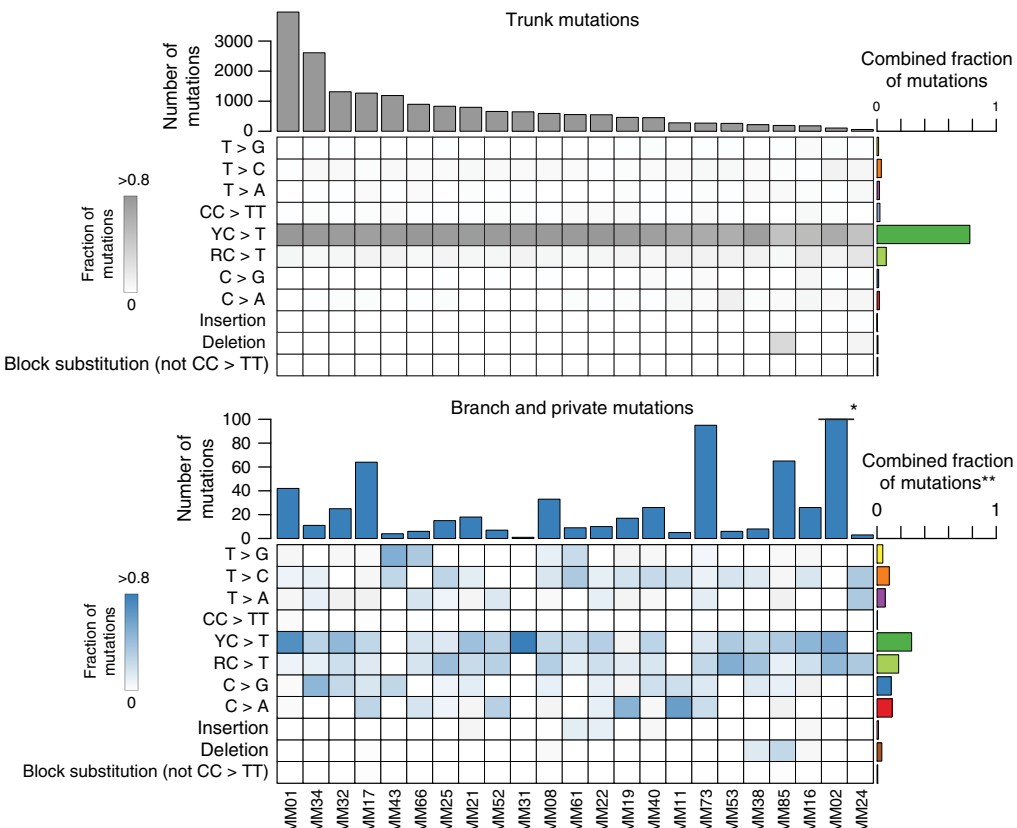

**Fig. 3** Comparison of trunk and branch mutations. Heatmaps show the relative frequency of mutations among branch mutations (top panel; gray) and branch mutations (bottom panel; blue) for each patient. C>T transitions are categorized as occurring downstream of pyrimidines (Y) or purines (R). The combined fractions of mutations represent the sum of mutations for each type relative to the total number of mutations for either trunk or branch mutations. * Due to the high number of branch mutations in MM02 ($n = 1786$), the bar is truncated for this patient. ** In the summary of branch mutation types, mutations in MM02 are omitted for clarity (branch mutations in MM02 displayed a particular mutational signature; see Supplementary Fig. 7 and 11 for details)

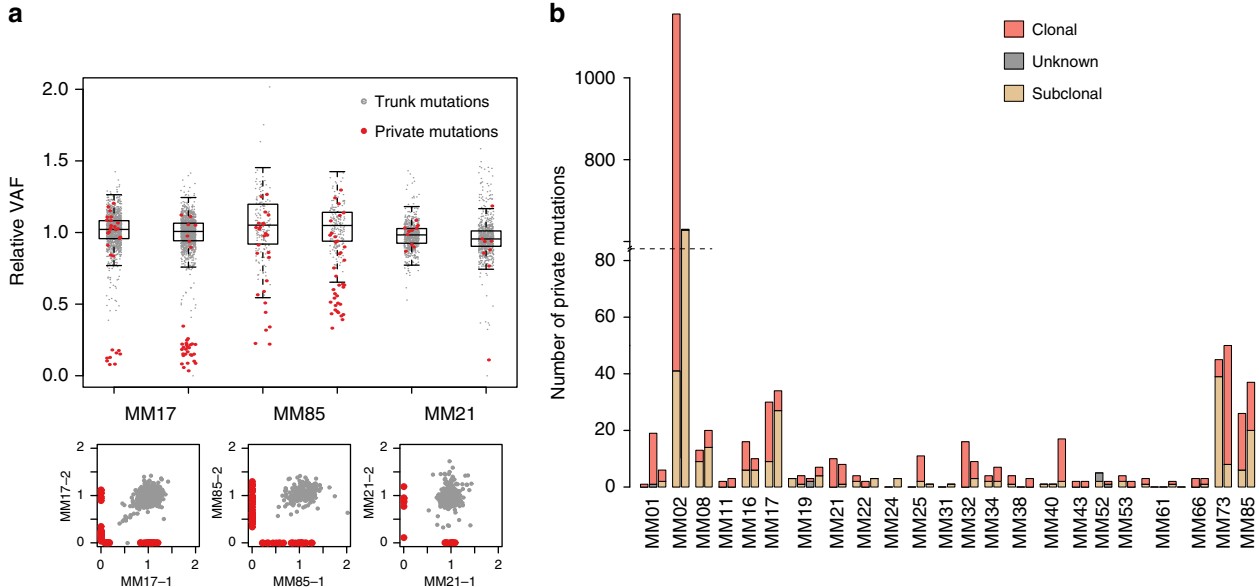

**Fig. 4** Cellular prevalence of mutations. **a** Relative variant allele frequency (rVAF); that is observed variant allele frequency corrected by tumor purity, local total copy number and estimated number of mutated alleles, for mutations in six representative samples from three patients. In theory, relative VAF is equivalent to cellular prevalence of the mutations. Mutations are colored according to presence in other lesions; gray, trunk mutations; red, private mutations. Boxes with whiskers are based solely on the trunk mutation relative VAFs and span the interquartile range (IQR), with whiskers extending to 1.5 times the IQR of the trunk relative VAF from the upper and lower bounds of the boxes. The three lower panels show examples of pairwise comparisons. **b** Private mutations were classified according to status as clonal or subclonal, where subclonal mutations are those whose relative VAF is below the whiskers in **a**. Mutations below half the median relative VAF and above the subclonality threshold are defined as unknown

one patient (MM61; Supplementary Figure 10), from whom five lesions were sampled, three were without clonal private mutations, and a number of shared mutations with low rVAFs were detected in multiple samples, possibly indicating a population of cells shared subclonally between lesions[31]. These findings may indicate polyclonal origins of, or reseeding between, lesions in this patient.

The common finding of private clonal mutations is consistent with a monoclonal origin of most metastatic lesions and indicates branching evolution. Furthermore, the observation of a UV-related mutational signature in a fraction of branch mutations (Supplementary Figure 7b) could indicate that different metastases may originate from different subclones in the primary tumor.

**Potential influence of therapy**. Two patients revealed atypical mutational patterns probably caused by prior therapy. One patient (MM02) had received two cycles of dacarbazine after mistakenly being diagnosed with metastatic disease. Eight months later she was correctly diagnosed with a distant subcutaneous metastasis to the abdominal region and a locoregional relapse, both of which were sampled. Nearly all private mutations were observed to occur clonally (within all cells) in the distant metastasis, but in a minor subpopulation of cells (~15%) in the locoregional relapse (Figure 11a) and were further attributed to a mutational process previously ascribed to temozolomide treatment in glioblastoma and melanoma[32,33]. Emergence of this signature has been found to depend on concomitant inactivation of DNA mismatch repair and, potentially, DNA methyltransferase *MGMT* in glioblastoma[33,34]. Here we identified three private *MSH6* mutations in two lesions sampled from this patient, all of which coincided with the (sub-)clonal populations of hypermutated cells (Supplementary Figure 11a). Further, reassessing previously published data[16], we identified transcriptional loss of *MGMT* in one, while the second sample revealed an *MGMT*

expression level close to the median across the sample set (Supplementary Figure. 11b). Notably, neither this signature nor mutations affecting *MSH6* was detected in tumors from any other of the three patients with at least one sample collected ≥6 months after dacarbazine therapy.

The second patient (MM85) received regional radiotherapy following surgical removal of a submandibular lymph node metastasis, with subsequent sampling of two metastatic lesions: a liver deposit (5 months later) and a subcutaneous lesion on the chest wall (6 months later; Supplementary Figure. 12a). Here, a large fraction of both trunk and private mutations constituted a unique mutational signature of small deletions, typically two nucleotides in length (Supplementary Figure. 12b), akin to a recently described pattern of mutations in radiation-induced secondary malignancies[35]. To the best of our knowledge, such a signature has not been described in melanomas. Strikingly, all private deletions were clonal, contrasting other private mutations in these samples (Supplementary Figure 12c). The finding of this signature in both subsequent samples located well outside the radiation field strongly favors the hypothesis of secondary spread, indicating the cells from the radiated submandibular area, and not the calvarian primary lesion (Supplementary Figure. 12a), to be the most recent common ancestor. However, in another five patients having tumor samples collected ≥6 months after initiation of radiation therapy, we did not observe a similar mutational signature. While we could not detect any mutations in DNA repair genes in the tumor tissues of patient MM85, it remains likely that this tumor may harbor particular defects conducive to the development of signature mutations in response to ionizing radiation.

**Sequence of genetic alterations during melanoma development**. The relative timing of genomic events occurring throughout cancer progression may be inferred by integrating information about copy number alterations and somatic VAFs[36,37]. The

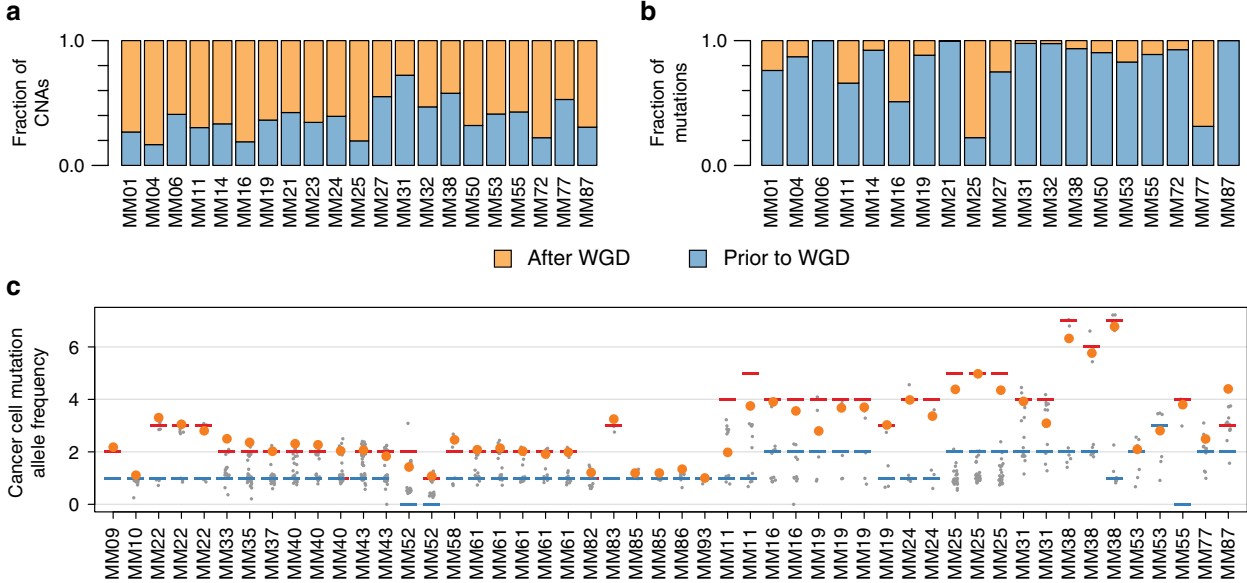

**Fig. 5** Timing of genome duplication and gain of mutated *BRAF*. **a**, **b** Estimated fraction of copy number events (**a**) and mutations (**b**) in assessable regions of the genome that occurred prior to and after genome duplication. **c** The relative allelic frequency (corrected to show allelic status of mutation) for mutations in the genomic segment harboring the *BRAF* gene in samples with a *BRAF* mutation. Allelic states are shown as red (major allele) and blue (minor allele) lines; mutations are shown as points, where the *BRAF* mutation is colored orange and all others are colored gray

finding of a higher genomic complexity (Supplementary Figure. 2c) and a higher copy number diversity (Fig. 2b) among patients with WGD is consistent with ongoing genomic evolution following WGD[38]. Indeed, the majority of copy number events in patients with WGD was estimated to have occurred after genome duplication ($n = 21$, median: 63%, range: 27–83%; Fig. 5a). Contrasting copy number alterations, most SNVs and indels appeared prior to WGD in most patients (median: 89%; range: 22–100%; Fig. 5b).

*BRAF* mutations are known to be early events in melanoma[39] and have been associated with an increase in *BRAF* copy number[5,22,40,41]. We observed low-level copy number gains of at least one *BRAF*-containing allele in 21/27 tumors with *BRAF* mutations, compared to four out of 26 in tumors wild type for *BRAF* ($p < 0.001$, Fisher exact test). The copy number gains all comprised broad regions of chromosome 7, except for a single patient harboring a focal (although still low level) gain of the *BRAF* gene. Strikingly, out of the 21 patients with concurrent mutation and copy number increase of *BRAF*, the mutated allele was the one gained in 20 patients (Fig. 5c). We did not observe associations between copy number elevations and driver mutations for any other oncogene, including *NRAS* (Supplementary Figure. 13). Interestingly, when assessing the allele-specific copy numbers of segments carrying *BRAF*, the most parsimonious solution indicated that *BRAF* gains are most likely to occur prior to WGD in eight out of nine informative patients.

Based on the evidence presented, we may postulate a general model for the order of events in the evolution of metastatic melanoma (Fig. 6). This model is characterized by early acquisition of driver mutations in key genes such as *BRAF* and *NRAS* which, in the case of *BRAF*, is usually followed by a gain of the mutated allele. Whole-genome duplication in general occurs as a later event, taking place after most UV-induced mutations, but prior to most copy number alterations. Following divergence of metastases, mutational accumulation is low and shifts away from UV-induced mutations to others, with a fairly consistent mutational rate within each patient.

## Discussion

While previous studies have described genomic alterations occurring in melanoma progression[8,42], including regionally advanced disease[9,43,44], limited knowledge exists in regard to distant metastases. To the best of our knowledge, this is the first study systematically exploring genomic heterogeneity in melanoma across multiple distant metastatic deposits.

We found most mutations to be truncal events. This is of relevance to driver mutations in particular, as we found a very low number of these to be heterogeneous in line with observations in regional metastatic disease[9,43,44]. The low number of heterogeneous mutations indicate metastatic divergence to be a late event, resembling recent findings in breast cancer[45]. Taking the observation of a UV-related mutational signature among some branch mutations into account, these findings are consistent with the hypothesis that different metastases may arise from separate late-developing subclones in the primary tumor, although other explanations may not be excluded.

We observed a surprisingly high intrapatient consistency regarding the number of private mutations across individual lesions. Interestingly, a similar phenomenon was recently described in metastatic breast cancer[45]. The finding of this phenomenon across two tumor forms with quite different mutational patterns[32] indicates this to be an intrinsic propensity related to several cancer forms. Moreover, the observation that heterogeneity correlates to *BRAF* mutation status, as was also made by others in primary melanoma[8], further supports the underlying genetic mechanisms associated with this process.

Our data indicate most metastases to have a monoclonal origin, even though we found indications of reseeding in one patient. This somewhat contrasts the findings of Sanborn and colleagues[9], who described reseeding as a more common phenomenon. Notably, many of the tumors from which they uncover shared subclones were locoregional relapses located in close anatomical proximity. Thick and large primary cutaneous melanomas are known to be associated with a substantial risk of locoregional relapse, despite wide margins in surgical excisions[46,47], consistent with local invasion, and it is reasonable to postulate that similar

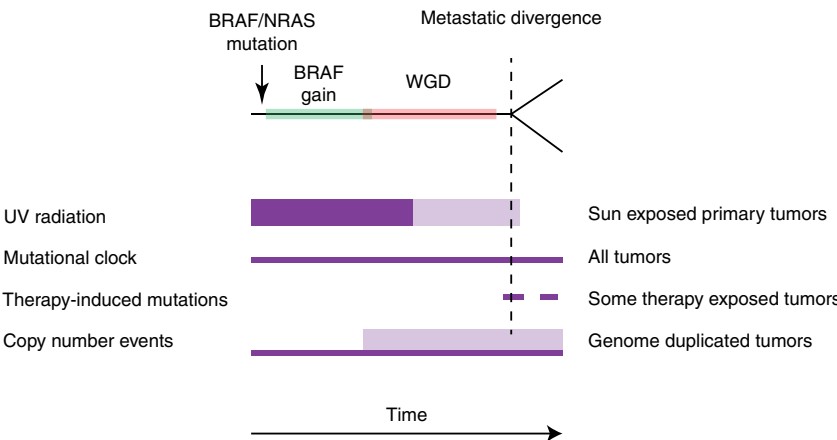

**Fig. 6** Model of progression for metastatic melanoma. Purple fields portray the timing of mutational processes, with increased thickness indicating higher mutational activity. Lower opacity indicates variability in timing of processes in relation to each other; e.g., timing of UV radiation in relation to the timing of genome duplication events

processes may regulate the development of locoregional relapses as well.

The patient (MM61) in whom we found indications of reseeding between metastases had an unusual clinical phenotype with numerous (>100 prior to death) cutaneous metastatic deposits on the truncus, shoulders and head area in addition to the 5 lesions sampled (Supplementary Figure. 10c-d). This suggests this cancer to have an organ-specific propensity for the development of cutaneous metastases[48], potentially, including a high migratory potential for metastatic cancer cells within the skin. Thus, while this patient presented with distant metastatic disease, the trafficking of tumor cells might be more akin to the pattern of reseeding observed in regionally disseminated disease[9].

An important topic relates to the sequence of genomic events during cancer progression. We found WGD to occur prior to metastatic divergence, and the high copy number diversity associated with WGD relative to near-diploid tumors suggests an ongoing process of copy number alterations, resembling findings in other tumor forms[45,49]. While genomic complexity is a classic prognostic marker in many tumor forms, how WGD relates to melanoma prognosis remains to be elucidated in larger series.

We found selective low-level gains of the mutated *BRAF* allele as a remarkably common early event in *BRAF*-mutated tumors, generally pre-dating WGD. *BRAF* mutations have previously been described in primary melanoma to be associated with the frequently observed arm- or chromosome-spanning gains of 7q[40], which is consistent with our current results. This likely contrasts *BRAF* gains associated with acquired resistance to BRAF-inhibitors which, when reported, has occurred through focal amplification of smaller segments[10,50]. While it seems reasonable to postulate low-level gain of *BRAF* to provide a selective growth advantage analogous to the fitness-gains associated with low-level gains of mutant *KRAS* in lung cancer[51], this issue warrants more research.

Emergence of the alkylating chemotherapy signature we observed in one patient has been related to DNA mismatch repair defects, with less evidence implicating inactivation of *MGMT* in glioblastoma[33,34,52]. While the signature has been described in melanomas subsequent to temozolomide treatment[32], so far it has not been related to any genomic alterations. Our findings of this signature in a patient harboring several *MSH6* mutations, but not among dacarbazine-exposed patients without mutations, may indicate DNA mismatch repair defects to play a role in melanoma as well.

Ionizing radiation is a well-known carcinogen[53], and secondary cancers arising in areas of previous radiation have been

described to reveal a distinct radiation-related mutational signature characterized by an accumulation of small deletions[35]. We found multiple private and truncal 2-nt deletions resembling this pattern of mutations in two distant metastatic deposits 5 and 6 months after radiotherapy for a regional lymph node metastasis. The issue of secondary metastatic spread remains controversial in melanoma[7], as well as in other tumor forms, much due to the fact that it is difficult to find direct evidence for this phenomenon. Chemotherapy exposure should affect tumor cells, including micrometastases, independent of anatomical location; in contrast, radiotherapy is applied to a localized area, with limited radiation scattering outside the treatment field. In this case, we found the radiation signature to constitute a form of "cellular labeling", strongly indicating secondary seeding from the radiation-treated lymph node to the chest wall and liver. While the biological effects of these radiation-induced deletions are unknown, the rapid emergence of two novel deposits <6 months after radiation both characterized by clonal 2-nt deletions should raise concerns that radiation therapy in some cases may enhance metastatic propensity and tumor aggressiveness.

In conclusion, this study provides evidence for common patterns of genomic alterations in melanoma progression. In most cases metastatic deposits seems to have a monoclonal origin with the possible exception of patients harboring multiple cutaneous deposits. The issue of potential secondary spread from metastatic deposits may have significant clinical implications; thus, further studies characterizing melanoma as well as other cancer metastases should seek to identify radiation-induced mutation signatures in all patients having previous exposure to radiotherapy.

## Methods

**Patients and sample collection**. The patients analyzed in this study were part of a single-arm prospective study assessing the response to dacarbazine therapy for metastatic melanoma[16,17]. Out of a total study population of 85 patients, 114 samples from 60 patients and corresponding benign tissue material (blood) were available for analysis by whole-exome sequencing. Samples from all biopsies were examined by a pathologist to ensure representative tissue. Data from 53 individuals (86 samples) are presented; the remaining samples were excluded due to low tumor cell content (<20%). Patient- and sample-level characteristics are detailed in Supplementary Tables 2 and 3, respectively.

All tumor samples were snap-frozen in the operating theater. Peripheral blood was collected at initial biopsy collection.

**Ethical approval**. The clinical study as well as the genomic analysis was approved by the Regional Ethics Committee of Western Norway (REK Vest; reference

numbers 020/00-109.99, 030/06-06/5520, and 2012/1740). All patients provided written informed consent.

**DNA sequencing**. Approximately 1 mg of genomic DNA from tumor and matched normal tissue were used for library construction using the Agilent SureSelectXT Human All Exon V5 kit (covering 50 mega-bases of exonic sequence). Libraries were paired-end sequenced using Illumina's TruSeq SBS chemistry v3 on a HiSeq2500, resulting in a median depth of coverage in the targeted regions ranging from 140 to 422 for tumor samples (median across samples: 271), and 43–233 for normal samples (median across patients: 87).

**Somatic variant calling pipeline**. Reads of each sample were mapped (lane-wise) with BWA mem[54] to the human reference genome (build b37 with an added decoy contig, obtained from the GATK resource bundle). Sample-wise sorting and duplicate marking was performed on the initial alignments with Picard tools (http://broadinstitute.github.io/picard). GATK tools[55] were subsequently used for two-step local realignment around indels, with matching samples (i.e., tumor and its corresponding normal) being processed together. Each sample's pair-end read information was then checked for inconsistencies with Picard and base-quality recalibration was performed by GATK. Somatic variant calling on the matching paired samples was done by using the intersection of MuTect[56] (somatic SNV detection) and Strelka[57] (somatic SNV and indel detection). Block substitutions were defined as somatic mutations at consecutive positions where the variant allelic frequency of each was within 5% of the average allelic frequency of the two variants. The program FastQC (http://www.bioinformatics.babraham.ac.uk/projects/fastqc/) was used for quality control of analysis input data. GATK tools were used for computing coverage statistics based on the recalibrated alignment files. Functional annotation of SNVs and InDels was performed with ANNOVAR (release 2015Mar22), using RefSeq as the gene transcript reference.

Most of the analysis (starting with the local realignment step) was limited to exome regions (the "exome" was in this context defined by Agilent exome v. 5 sequencing probe targets).

**Driver mutation definitions**. Mutations in a set of genes previously identified as drivers in melanoma[3,21,22] were manually assessed for likely status as drivers. For all considered genes, driver mutations were defined as drivers if they (1) were canonical melanoma-associated mutations; (2) as likely drivers based on evidence of gain or loss of function in the published literature, or if the positions were recurrently mutated in other forms of cancer; or as (3) inactivating if they occurred in tumor suppressors and disrupted the protein reading frame (i.e., nonsense, frameshift, or splice site mutations). Otherwise, mutations were deemed to be passengers (Supplementary Table 4). Patients were categorized according to driver mutation status in *BRAF*, *NRAS*, and *NF1*, where mutations at the canonical mutational hot-spots for *BRAF* (p.V600 or p.K601) and *NRAS* were prioritized in the case of driver mutations in more than one of these genes.

**Mutational signature analysis**. DeconstructSigs[26] was used to estimate the contribution of mutational processes to the observed patterns of mutations. Contributions from 5 mutational processes that have been described in melanoma were assessed (signatures 1, 5, 7, 11, and 17)[32,58]. Observed mutational patterns were corrected for the 3-base composition of exonic regions in the genome. Signatures reported in the COSMIC database (v79) were used as reference for the mutational pattern associated with each process[58].

For signature analysis of branch mutations we used a lower threshold of $n = 10$ mutations. Given that this number of mutations is too low for precise estimates of percentage contribution to individual signatures, we also performed manual assessment of mutations, focusing on typically UV-related mutations (such as YC>T transitions).

**Copy number profiling**. Copy number profiling was performed using an in-house algorithm optimized for the present dataset. Our algorithm was established to take advantage of two features in the data:

1.  To optimize CNA and tumor purity estimates by use of the observed variant allele frequency of somatic mutations (i.e., to fit CNA estimates on to VAF of SNVs).
2.  In the cases with multiple samples per patient, to take advantage of samples with high tumor purity to optimize allele-specific copy numbers across samples within the same patient.

In brief, copy number determination was carried out in three stages: First, segmentation was performed based on shifts in observed allele frequencies of heterozygous SNPs between genomic regions with differences in copy numbers. Second, allele-specific copy numbers across the genome, as well as tumor cell content, were estimated based on the magnitude of shifts in allele frequency of heterozygous SNPs relative to regions with a loss of the minor allele, or based on allele frequency of somatic mutations in the absence of copy number alterations. Third, in patients with multiple samples, cross-sample corrections were made for breakpoint identification and copy number determination based on a combination of germ line and somatic variant allele frequencies. False discovery rates were

estimated by simulation, rather than SNP-based benchmarking tools[59], since the current dataset was restricted to WES.

The algorithm was based on the allelic frequency of germ-line variants in tumor and normal samples. Based on the ratio of sequencing depth between tumor and normal, tumor allelic copy numbers uncorrected for normal cell content, the relative copy number (RCN), can be observed. In theory, the interval between RCNs is directly proportional to the difference in number of alleles between adjacent copy number segments. Therefore, the absolute tumor copy number (TCNs) can be determined through inferring the interval of a RCN and the lowest observed RCN value, which normally corresponds to a copy number of zero, or loss of one allele. Based on this, we performed copy number profiling, as follows:

**Segmentation**. Identification of potential breakpoints: Potential break points were identified based on shifts in allelic frequency of heterozygous SNPs in each tumor relative to the corresponding normal sample across chromosomes. Here, a sliding window approach was used, where the genome was split into bins of 4 Mb, with a step size of 1 Mb. If the number of SNPs in a given bin was <40, the bin was merged with the nearest neighboring bin. For the $i$-th bin, which included $k_i$ SNPs, we compared the standard deviation of major allele frequency between tumor and normal sample. If there was no difference, the B allele frequency (BAF) of the bin was regarded as 0.5 ($b_i = 0.5$). Otherwise, $b_i$ was defined as the median value of allele frequencies $m_i$. A potential break point containing region was defined by a difference in BAF exceeding 0.015 between adjacent bins. This cutoff at 0.015 was determined by simulation of randomly generated break points:

At each BAF ranging from 0 to 1, with increments of 0.01, we generated 1000 simulation datasets, each including 40 segments. A randomly assigned number of SNPs was assigned to each segment, ranging from 40 to 1200, and coverage of each SNP followed the distribution of SNPs in the current exome sequencing dataset (geometric distribution; $p = 0.01$). Allelic read counts were modeled using the binomial distribution B(N, BAF), where N was the total sequencing depth of the SNP, and BAF ranged from 0 to 1. For the simulation data corresponding to each BAF we calculated the absolute differences of average BAF from all SNPs between two adjacent segments. In order to determine the significance of difference, we defined an empirical *p*-value for the likelihood that two segments corresponding to the same theoretical BAF were randomly separated. We estimated the empirical *p*-value based on the simulation data corresponding to each BAF, and found that a difference of 0.015 corresponded to an empirical *p*-value of 0.05.

$$p = \frac{\#(\Delta \geq \text{cutoff})_{\text{BAF}}}{(\#(\text{segments}) - 1) \cdot \#(\text{simulation})},$$

where # represented the counts and $\Delta$ represented the difference between adjacent segments.

Determining the precise breakpoint and merging of segments: In order to determine a more precise breakpoint between bins, regions flanking the potential breakpoint (±4 Mb) were split into smaller windows, each including 3 SNPs. For each SNP, we used the major allele frequency ($m$) in the following analysis. The average $m$ value of the first window was compared to the rest of the flanking region. If the difference was more than 0.018, the midpoint of these two sub-regions was regarded as the final breakpoint. The cutoff at 0.018 was determined by estimation of simulation data using the same parameters as above. Based on the simulation data, we found the maximum random error between adjacent segments with the same theoretical BAF was no more than 0.018; although, the random error increased when BAF was closer to 0.5 (Supplementary Figure 14). If a difference of more than 0.018 was not identified by this initial assessment, the window was extended to encompass the second window and compared to the rest of the flaking region. This procedure was repeated until a break point was found, or until the end of the flanking region was reached. If no break points exceeding 0.018 were found, the potential break point was discarded. The genome was thus split into multiple segments according to these final break points, and $m$ values corresponding to each segment were estimated ($m^i$ and $\overrightarrow{m_i}$ representing the major allele frequency of segment $i$ and maximum allele frequency of all SNPs in segment $i$, respectively).

**Estimating allelic tumor copy numbers and tumor purity**. The relative copy number (RCN) of each allele for each segment can be obtained based on the following formula.

$$\text{CNAH}_i = m^i \times \text{ratio}_i,$$

$$\text{CNAL}_i = (1 - m^i) \times \text{ratio}_i,$$

where $i$ was the $i$-th segment. CNAH represents the relative allelic copy number of the major allele, and CNAL represents the relative allelic copy number of the minor allele. Ratio represents the ratio of sequencing depth between tumor and control. For each segment, we estimated the allelic RCNs based on the CNAH and CNAL.

CNALs of all segments were integrated by multiplying it with the number of SNPs in each segment. If the resulting distribution of weighted relative minor allele copy numbers had at least two peaks, we considered the minimal peak value as CN

= 0, and the second as CN = 1 or CN = 2. The distance between copy numbers, DIS, can be calculated based on CN = 0 and CN = 1 or CN = 2.

$$\text{DIS} = \text{CN1} - \text{CN0}$$
$$\text{DIS} = (\text{CN2} - \text{CN0})/2.$$

According to the CN0 and DIS values, allelic TCN of each segment corresponding to each relative copy number state was determined as follows:

$$\text{dCNA}_i = \text{round}\left(\frac{\text{CN}_i - \text{CN0}}{\text{DIS}}\right).$$

Where $i$ represented the $i$-th segment and round means that values are rounded to the nearest nonnegative whole number. Major (CNH) and minor (CNL) allelic copy numbers were thus calculated.

A tumor sample consists of a mixture of tumor cells and normal cells. For each locus in a chromosome, the expected major allele frequency of SNPs can be calculated as follows:

$$f_{\text{major}} = \frac{\alpha \times \text{CNH} + (1 - \alpha)}{\alpha \times \text{CN}_{\text{total}} + 2(1 - \alpha)},$$

where $\alpha$ is the tumor cell fraction. Thus, the tumor purity for each imbalanced segment was estimated through the following formula.

$$\alpha = \frac{2 \times f_{\text{major}} - 1}{\text{CNH} - 1 + f_{\text{major}}(2 - \text{CN}_{\text{total}})}.$$

Weighting genomic segments as above, the density peak of purities calculated across segments was used as the tumor purity for each sample. Without sufficient imbalanced segments to obtain tumor purity, we used mutation allelic frequencies in balanced segments (TCN = 2; i.e., mutations on one allele) to estimate tumor purity. The allelic frequency of mutations is given by

$$f_{\text{mut}} = \frac{\alpha \times \text{CN}_{\text{mut}}}{\alpha \times \text{CN}_{\text{total}} + 2(1 - \alpha)}.$$

Thus, the tumor purity would be the local peak of mutations density across genome segments with balanced copy number 2. The tumor purity was then inferred from the mutation allele frequency in these segments:

$$\alpha = 2 \times f_{\text{mut}}.$$

**Inferring allelic tumor copy number and tumor purity of non-reference sample**. For patients from whom multiple samples were analyzed, data from the different samples was used to adjust each other, adding strength to the estimates. In these cases, the sample with the highest tumor purity was coined the "reference sample" while the others were termed "non-reference" samples. Segments in non-reference samples with copy number 0 or copy number 1 were inferred from the corresponding segments in the reference sample. The difference in relative copy numbers (DIS) was estimated based on these segments with copy number 0 and copy number 1. Allele-specific copy numbers were re-evaluated based on the CN0 and DIS estimates, and tumor purity was calculated as shown above. For each non-reference sample, if TCNs of 50% segments differed from the reference sample, we would re-infer the TCN for this non-reference sample in case of genome doubling or tripling.

**Estimation of multi-sample tumor allelic copy numbers by clustering of somatic mutations**. The estimation of TCNs based on frequencies of mutations can be used to tune the accuracy of copy number calls estimated from SNPs. This approach can be strengthened by use of multiple samples from the same patient. In the present study, such additional tuning was performed for patient MM01, due to the combination of low tumor cell fraction and high ploidy. Thus, we submitted mutations shared between different samples from this patient to K-mean clustering based on variant allele frequencies of mutations in all combinations of the patient's samples. The number of clusters, $k$, was defined to select the optimal clustering. Here, the number of clusters resulting in the minimum average sum of squared errors $E(C)$ for $k$ in the range of 2–5 was selected, where $E(C)$ was defined as:

$$E(C) = \frac{\sum_{s=1}^{\binom{n}{2}} \sum_{t=1}^{k} \sum_{o \in C_{ts}} d(o, \text{cen}_{ts})}{\binom{n}{2}},$$

where $n$ was the number of samples, $s$ was the combination of two sample, $\text{cen}_{ts}$ was the centroids of cluster $t$ in combination $s$, and $o$ represented the mutation in cluster $t$ of combination $s$. The distance $d$ was calculated as Euclidean distance.

The optimal combination of pairwise comparisons of samples based on clustering was regarded as a standard to infer TCNs of each sample from the same patient. For each previously identified segment of the samples, the median value of mutation allele frequencies, corrected for copy number and tumor cell content, mapping into each standard cluster was regarded as the value of the cluster. We determined the optimal combination of two samples based on maximization of inter-cluster distances and minimization of intra-cluster distances. First, the distance between clusters from a combination of two samples was calculated. The distance $d$ between two clusters $C_i$ and $C_j$ was defined as the Euclidean distance between the cluster centroids $\text{cen}_i$ and $\text{cen}_j$.

$$d = \sum_{i=j} d\left(C_i, C_j\right) = \sum_{i=j} d\left(\text{cen}_i, \text{cen}_j\right).$$

The combination with maximum clustering distance was retained. In cases with more than one possible combination, the optimal combination of two samples was derived from the minimum average intra-cluster distances between centroids; the intra-cluster distance being defined as:

$$d = \frac{\sum_{t=1}^{k} \sum_{o_i \neq o_j \in C_t} d\left(o_i, o_j\right)}{\sum_{t=1}^{k} \binom{|C_t|}{2}},$$

where $|C_t|$ was the number of mutation cluster $C_t$. The combination with the minimum intra-cluster distance was regarded as the optimal combination of two samples.

Mutation frequencies of all standard clusters from all segments in the sample were integrated to estimate their probability densities. For any tumor copy number (TCN) state, $F$, local peak values of mutation frequency distributions were regarded to correspond to specific copy number states, $f$.

$$F = (f_1, f_2, \ldots, f_n)$$

$$f_1 < f_2 < \ldots < f_n,$$

where $f_i$ was the $i$-th local peak in mutation frequency distribution. The minimum mutation frequency ($f_i$) in $F$ was defined as corresponding to copy number of 1. We calculated the interval of TCN as the difference between each $f_i$ and $f_{i+1}$. Further, based on $f_1$ and interval of TCN, CNH, and CNL of each segment in the sample were obtained.

**Estimation of false discovery rates**. To estimate the false positive CNA calls corresponding to the applied cutoff (a difference in BAF of 0.018 between segments), we assumed scenarios where the total copy number in tumor cells ranged from 1 to 8 following a uniform distribution. We simulated 1000 segments (similar with previous simulation process) under different tumor purities ranging from 1 to 100%, with the different total copy numbers (1–8, respectively; Supplementary Figure 15). Based on the segments with the same BAF, combining all tumor purity and total copy numbers, we found the global average false positive rate (FPR) to be 9.88% and the global average false negative rate (FNR) to be 8.44%. The FPR and FNR decreased with the increasing of tumor purity. At tumor purities below 20%, FPR and FNR increased rapidly. Importantly, when the tumor purity was higher than 20%, FPR and FNR was always <10% (Supplementary Figure 15).

**Exclusion of samples from analysis**. Simulations (see above) introducing different percentages of reads from normal DNA into samples of data from tumor DNA, indicated that aberrant cell fractions higher than 20% was sufficient for accurately calling copy number alterations. Out of the 114 tumor samples that underwent sequencing, 86 fulfilled this criterion and were used in subsequent analyses.

**Inference of whole-genome duplication**. For each sample, to infer whether a whole-genome duplication event had taken place, we enumerated the fraction of the genome with a minor allele at copy number 2 and the estimated ploidy. A manual assignment was then performed, based on the assumptions that (1) the overall ploidy of a sample having undergone genome duplication would generally be higher than those of diploid samples, and (2) that the minor allele should be at copy number 2 in at least some fraction of the genome after a whole-genome duplication event (Supplementary Figure 2b).

**Mutational heterogeneity between samples**. For the analysis of inter-lesional mutational heterogeneity, we considered only mutations whose heterogeneity could not be reasonably be explained by copy number alterations or lack of sequencing depth. Thus, mutations were considered to be potentially heterogeneous if (1) in a sample without a particular mutation, there was no evidence of copy number loss relative to samples carrying the mutation; and (2) the sequencing depth at the position was high enough to have a 95% chance of detecting the mutation given an

allelic fraction of 1 allele out of 4 and the sample-specific tumor cell fraction, assuming a binomial distribution of variant reads. This resulted in a sample-wise depth threshold ranging from 56 for samples with a low aberrant cell fraction, to 18 for samples with a high aberrant cell fraction. In addition, a mutation that was not called by the somatic variant calling pipeline was deemed to be present if the number of reads supporting the mutation was over 1 and higher than what would be expected with an error rate of 1/200, assuming a binomial distribution of supporting reads, with a binomial test $p$-value of under 0.05. One patient (MM43) exhibited parallel loss of chromosomes 11q and 14 in each of the sampled lesions. Heterogeneous mutations on these chromosomes were considered to have been lost due to copy number alterations.

**Calculation of relative VAF and assessment of clonality**. As a measure of the cellular prevalence of each mutation, we calculated the relative variant allele frequency (rVAF) of each mutation as the ratio of observed to expected VAF, given local copy number state, tumor cell content and estimated number of mutated alleles:[37]

$$\text{rVAF} = \frac{\text{VAF}_{\text{obs}}}{\text{VAF}_{\text{exp}}} = \frac{\text{VAF}_{\text{obs}}}{\left(\frac{n_{\text{mut}} \times \rho}{2 \times (1-\rho) + n_{\text{tot}} \times \rho}\right)},$$

where $n_{\text{mut}}$ refers to the number of mutated alleles, $n_{\text{tot}}$ refers to the total copy number at the mutated locus, and $\rho$ refers to the tumor cell content.

Relying on the accuracy of the determination of inter-lesional mutational heterogeneity, we evaluated the clonality of mutations by comparing the rVAF of trunk mutations to that of private mutations to infer likely clonal relationships, using clustering of mutations across samples to validate our findings[31]. Evaluations of mutation clonality were based on the interquartile range (IQR) of rVAF values of trunk mutations only. Thus, mutations were categorized as being subclonal if their rVAF were below the 25th percentile by 1.5 times the IQR, and otherwise as clonal if their rVAFs were above 0.5 times the median rVAF. Mutations not specified as subclonal, and with rVAFs below 0.5 times the median rVAF were considered to be of unknown clonality.

**Relative timing of whole-genome duplication**. To determine the fraction of copy number events that preceded or followed genome duplication, the shortest route to obtain the observed copy number state for each segment was determined. Here, a copy number change before duplication would lead to a change in observed copy number of two copies from the "unaltered" state of two copies, and a copy number change after genome duplication would lead to a change of one copy. Solving the resulting equation for the minimum number of events, the sum of events occurring prior to and following genome duplication was estimated for each allele in each segment. For each patient, the average number of events across samples was used as a measure of copy number changes prior to and following duplication. To estimate the number of mutations that occurred prior to and following genome duplication, mutations at each allelic state in informative regions of the genome (those with major:minor allele states of 2:2, 2:1 or 2:0) were enumerated. The fraction, $m_1$, of mutations preceding duplication was estimated as $m_1 = \frac{3n_2}{n_1 - n_2}$ for copy number 2:1, or $m_1 = \frac{n_2}{2n_1}$ for copy number 2:2 and 2:0, where $n_1$, and $n_2$ were the number of mutations with allele status 1 and 2, respectively.

**Statistical analyses**. All statistical analyses were performed in the statistical programming language R (v3.4.1)[60]. Ranked tests were used for comparisons of continuous variables across groups (Mann–Whitney $U$-tests or Kruskal–Wallis rank-sum tests), or when assessing correlations between continuous variables (Spearman's rank correlation), except if otherwise specified. All significance tests were two-sided, and statistical significance was considered for $p < 0.05$.

**Data availability**. Raw sequencing data are not publicly available due to national regulations regarding privacy concerns of study participants. Data on somatic mutations are presented in Supplementary Table 1.

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

## Acknowledgements

Parts of this work were performed in the Mohn Cancer Research Laboratory. The work was funded by the Norwegian Research Council through the Norwegian Cancer Genomics Consortium (NCGC; grant numbers 218241 and 221580), The Norwegian Health Region West, The Norwegian Cancer Society and The Bergen Research Foundation. We thank Dagfinn Ekse for technical assistance.

## Author contributions

Patient recruitment: J.G. and P.E.L. Curation of clinical data: E.B., J.G., and P.E.L. Data analyses: E.B., S.Z., D.P., S.N., D.V., L.A.M.-Z., and E.H. Provided research infrastructure: O.M. and P.E.L. Study design: E.B., S.Z., J.G., S.K., and P.E.L. Manuscript writing: E.B., S. K., and P.E.L.

## Additional information

**Competing interests:** The authors declare no competing interests.

