## [Peer Review File · Nature Communications]

Reviewers' comments:

Reviewer #1 (Remarks to the Author):

The authors report coding region somatic substitutions, indels and copy number alterations for 86 metastatic melanoma samples that have arisen from the predominately cutaneous primary tumours of 53 cases, using exome sequencing. For 23 cases with multiple samples they report an overall pattern of heterogeneity of fewer branch variants as compared to trunk events and show that the inter-metastatic profile of mutations is largely stable for the majority of these cases (except in 2 cases). The 7 mutational signatures reported to be predominant in melanoma samples were used to fit the somatic variants in order to assess the UV signature contribution to mutations in metastases arising from sun exposed primaries. The relative timing of whole genome duplication events is also presented as a late event in most cases.

The strength of this study is that the description of the heterogeneity patterns identified in metastatic melanomas is novel and well timed with current interest in advanced disease genomics growing.

A major weakness concerns the potential reproducibility of the work on the basis of the lack of data demonstrating the sensitivity and specificity of the in-house copy number tool at different tumour purities and the approach used for the evolutionary clonal analysis.

The effectiveness of the in-house copy number tool underpins the main findings of this study. It is used to estimate tumour sample purity, call major copy number variants and identify tumour cell ploidy. It is reported to be able to accurately identify tumour cell content down to 20%, which is also the limit for copy number profiling. Due to biases inherent in exome sequencing from the capture through to alignment copy number determination and particularly identification of sample ploidy are very challenging. As sample purity decreases the ability to identify somatic copy number changes degrades especially for deletions.

Can the authors provide measures of sensitivity and specificity for their copy number tool that indicate its effectiveness particularly at 20% tumour cell content such as described in a recent review Zare et al. BMC Bioinformatics (2017) 18:286 DOI 10.1186/s12859-017-1705-x.

Additionally a summary of the estimated sample purity for each sample may help the reader judge the confidence of the copy number and ploidy calling.

The approach for defining the branch mutation as absent from some samples and trunk mutations as present in all samples does not determine the clonality of a mutation. By calculating the percentage of cells (corrected for local copy number tumour sample purity) in each sample that contains each mutation the clonal composition of the related samples will be revealed. This approach would also allow the clonal analysis to be extended to the single sample cases and provide a more robust analysis.

Minor omissions and typos

- Results section line 101 refers to “table 1” which is not present in the manuscript bundle.
- Supplementary information:
 - o Line 104 “duplicating” do you mean multiplying
 - o Line 105 “weighed” do you mean weighted
 - o Line 128 “weighing” do you mean weighted

Reviewer #2 (Remarks to the Author):

The authors have analysed 86 lesions from 53 metastatic melanoma patients to study the genomic evolution of metastatic progression in this cohort. Using high-depth WES (Whole Exome Sequencing) data and copy number profiling, the authors first examined the genomic complexity of the lesions, which they define as the fraction of the genome in an aberrant state, and conclude that samples in which whole genome duplication (WGD) is evident (approx. 40%), are more genomically complex. Additionally, patients whose tumours display evidence of WGD, display a larger copy number diversity, defined as the mean number of copy number alterations separating samples from individual patients, but overall patients displayed a low degree of mutational diversity. The authors go on to assess mutational processes, showing that trunk mutations are dominated by UV signature, polyclonal seeding and the influence of therapy in these samples, and propose a model for the progression of melanoma metastasis wherein the acquisition of driver mutations, such as BRAF, is usually followed by a gain in the mutated allele, and in which WGD is an event that happens mostly after the UV-induced mutation but before any copy number alteration.

The authors present an in-depth analysis of the genomic landscape of a cohort of melanoma metastases, and their findings are in keeping with previous studies. However, a number of issues would need to be addressed for their conclusions to be adequately supported:

- The authors have concluded, based on data from metastatic samples only, that, as they say in the abstract, metastases may arise from different subclones in the primary tumour. However, they have no evidence of this, given they have not analysed any primary tumour material to confirm this.. Their extrapolation of their data to positing theories about tumour evolution is somewhat problematic, as at best, they are only comparing intra-patient metastatic lesions, rather than primary to metastatic evolution
- Whilst the authors claim to present 86 lesions from 53 patients, these include a number of samples from acral, mucosal, and uveal melanoma, from which only a single lesion was analysed. The bulk of the study pertains to 23 patients from whom multiple lesions were obtained, and whilst the authors present some analysis of the single lesions in figure 1, they have not inferred any conclusions for these samples later in the rest of the study. In fact, the authors have not shown or made any conclusions about whether the evolution progression model proposed by them works for different melanoma types and single lesions. Having other types of melanomas in the cohort doesn't provide any novel insight.
- In the method section for copy number profiling, authors have extensively used the simulated data to tune parameters. Apart from the distribution type, the details on the simulated data are lacking. Further, in the estimating allelic tumor copy numbers and tumor purity section, equations do not have any prior algorithms referenced along with them. This makes it hard to comprehend how the equations were derived. Lastly, without the availability of the scripts/codes for copy number pipeline it becomes difficult to assess the advantages of this pipeline.
- Authors have performed mutational signature analysis separately for trunk and branch mutations. The signature analysis for branch mutation is affected by increased residual sum of squares error in model fitting when the numbers of branch mutations are less than 50 (as also shown in Fig S7). Hence the results are affected by mathematical uncertainty in parameter estimation. Further, as shown in the Figure S7 (b), only one patient has predominance of UV signature, hence the conclusion that some branch mutations fit into UV pattern is flawed.
-

Overall it seems to the reviewer that a key weakness of this study is the lack of evidence from primary tumour material to support their proposed model. Conclusions have been drawn on somewhat circumstantial evidence, providing little advancement in knowledge of metastatic melanoma evolution. The authors acknowledge that previous studies have described genomic alterations in melanoma progression, and claim to advance knowledge about distant metastases, but given there appears little difference between physically proximal and distant metastases in this study, the reviewer does not find the current study provides and advancement on these previous studies.

Minor points

At several places, tables from the main paper are missing and all the supplementary tables are not labelled.

- There are several typos and mismatched references for the figures and tables in the paper. For example in driver mutations and genomic complexity section it should have been Fig 1(c) instead of fig 1(d).

Reviewer #3 (Remarks to the Author):

The authors perform a comparative analysis of 86 distant metastatic lesions from 53 melanoma patients (multiple lesions available for 23/53). In order to characterize the genomic alterations underlying disease progression, they perform whole exome sequencing and identify coding mutations and copy number changes. Integrating the two data, they show that mutated BRAF allele is usually amplified before whole-genome duplication (WGD), which happens before metastatic spread. They also conclude what WGD is correlated with greater genomic complexity in comparison with those tumors without WGD. They characterize the subclonal structure and determine the relationship between different tumor sites from the same patient. They identify trunk as well as

branch mutations and report that all driver mutations are on the trunk. Comparing the variant allele fraction of private mutations to the mutations on the trunk, they conclude that there is no evidence of polyclonal seeding in these patients. Cohorts of this many patients with multiple metastatic samples available are rare and overall the analysis presented is sound and of interest to other researchers in the field of cancer genomics. However, there are some minor points that needs to be addressed:

1. To characterize subclonal structure, the authors first determine if a mutation is shared across all sites (trunk) or private to a particular sample (private). They correct variant allele frequency (VAF) for locus-specific copy number and tumor purity to calculate relative VAF and compare the shared and private mutations to identify private mutations that are clonal. They conclude that existence of private clonal mutations indicates monoclonal seeding. Although the inference of the monoclonality is probably true, the authors should perform a statistical clustering of relative VAF across different samples from the same patient to identify mutation clusters. There are many published methods to do this kind of analysis including PyClone that the authors say they chose not to use. What is the reason why the authors opted out of using such an algorithm?
2. Figure 2b is difficult to understand. The range of the y-axis for branch and trunk mutations are different. Representation of this information would be best done with phylogenetic trees. Alternatively the author could include a third panel with a bar plot showing the proportion of branch/trunk mutations per sample.
3. The information in Figure 4a should be plotted for each sample pair in a way similar to Figure S10a. This will clearly show that trunk mutations have relative VAF of 1 in the two samples under comparison and there are indeed clonal private mutations in most samples. If space is an issue, this could be provided as a supplementary figure.
4. For mutation signature analysis, a minimum of 10 mutations is not enough to perform reliable signature assignment. It might be better to leave this analysis out and only show the information summarized in Figure 3, which is already clearly indicating a change in mutational landscape in later subclones.
5. Did the authors interrogate the copy number changes for driver alterations?
6. Could supplementary table 1 be extended to include sample-level information such tumor purity etc.?

Response to reviewers' comments:

Reviewer #1:

The authors report coding region somatic substitutions, indels and copy number alterations for 86 metastatic melanoma samples that have arisen from the predominately cutaneous primary tumours of 53 cases, using exome sequencing. For 23 cases with multiple samples they report an overall pattern of heterogeneity of fewer branch variants as compared to trunk events and show that the inter-metastatic profile of mutations is largely stable for the majority of these cases (except in 2 cases). The 7 mutational signatures reported to be predominant in melanoma samples were used to fit the somatic variants in order to assess the UV signature contribution to mutations in metastases arising from sun exposed primaries. The relative timing of whole genome duplication events is also presented as a late event in most cases.

The strength of this study is that the description of the heterogeneity patterns identified in metastatic melanomas is novel and well timed with current interest in advanced disease genomics growing.

A major weakness concerns the potential reproducibility of the work on the basis of the lack of data demonstrating the sensitivity and specificity of the in-house copy number tool at different tumour purities and the approach used for the evolutionary clonal analysis.

The effectiveness of the in-house copy number tool underpins the main findings of this study. It is used to estimate tumour sample purity, call major copy number variants and identify tumour cell ploidy. It is reported to be able to accurately identify tumour cell content down to 20%, which is also the limit for copy number profiling. Due to biases inherent in exome sequencing from the capture through to alignment copy number determination and particularly identification of sample ploidy are very challenging. As sample purity decreases the ability to identify somatic copy number changes degrades especially for deletions.

Can the authors provide measures of sensitivity and specificity for their copy number tool that indicate its effectiveness particularly at 20% tumour cell content such as described in a recent review Zare et al. BMC Bioinformatics (2017) 18:286 DOI 10.1186/s12859-017-1705-x.

Response:

We understand that the rationale for using an in-house algorithm was not properly explained in our original manuscript and have now provided an explanation for this in the supplementary information. The rationale for employing an in-house algorithm for calling copy number alterations was two-fold:

- 1. We wished to optimize CNA and tumor purity estimates by use of the observed variant allele frequency of somatic mutations (i.e. to fit CNA estimates on to VAF of SNVs).*
- 2. In the cases with multiple samples per patient, we wished to take advantage of the sample with high tumor purity to infer allele specific copy numbers across samples within the same patient.*

We found that both of these points improved our accuracy in determining copy number states in samples with low tumor purity. Notably, analyzing our data set with established CNA-calling tools, such as ASCAT, we found apparent overcalling of segments. (For the Editor and the Reviewers' information, we attach a set of figures (figure Rev01) at the end of this response letter illustrating this point). We did not include these figures in our revised manuscript; however, in case the reviewers / editor would like to have it included in the Supplement, we of course will do so.

The reviewer requests us to follow the procedure used by Zare et al. for assessing the accuracy of our copy number tool, particularly for determining accuracy at lower tumor purities. This would, however, require benchmarking with complementary methods (i.e., SNP-arrays) which were not available for our samples. Also, simulations mimicking varying tumor purity were not available in the resources used by Zare et al. To answer the reviewer's request in an alternative manner, we assessed the ability of our tool to call changes in copy number under varying conditions by use of simulation data. Here, we incrementally increased the fraction of "normal cell" genomic information (i.e. CN=2) onto tumor sequencing data, and observed that the rates of both false positives (FPR) and false negatives (FNR) were low (<10%), for tumor purities >20%. The FPR and FNR only increased notably with tumor purities below 20%.

We realize that this information about the algorithm's performance should be made available for the reader and have now added extended descriptions of this in the Supplementary information. This includes the new section "Estimation of false discovery rates" and the relevant new figure (Supplementary figure S15) illustrating the copy number tool's performance under different conditions with respect to tumor purity.

Additionally a summary of the estimated sample purity for each sample may help the reader judge the confidence of the copy number and ploidy calling.

Response:

We concur with the reviewer's comment. We have thus included a new supplementary table (Supplementary table S3) containing sample-wise information including measures of tumor purity.

The approach for defining the branch mutation as absent from some samples and trunk mutations as present in all samples does not determine the clonality of a mutation. By calculating the percentage of cells (corrected for local copy number tumour sample purity) in each sample that contains each mutation the clonal composition of the related samples will be revealed. This approach would also allow the clonal analysis to be extended to the single sample cases and provide a more robust analysis.

Response:

We concur that the approach used to define mutations as present or absent does not assess the clonality of mutations when present. It does however take into account the most important technical and biological parameters, including the potential for

lack of sequencing depth and copy number loss in a region with a seemingly lost mutation. Clustering of the cellular prevalence to determine the clonal structure of mutations across samples would not add further information to the question of whether a mutation is present or not. Still, as the reviewer points out, the clonal composition of related samples may be assessed by such analyses. Our experience with mutation clustering algorithms, however, is that results from these in many cases lead to incompatible evolutionary trajectories for the samples under comparison (see also response to reviewer #3), where daughter clones in one sample may be inferred to be a parent clone in another. We were therefore inclined to apply such clustering algorithms with caution. However, since we do recognize the potential value of such analyses, we have now included the results of multi-sample clustering of mutations in supplementary figure S9, and have replaced figure S9a in the original manuscript with one based on analysis by PyClone (now figure S10a).

Minor omissions and typos

- Results section line 101 refers to “table 1” which is not present in the manuscript bundle.

Response:

By mistake, table 1 was not enclosed in the first submission. It is now enclosed.

- Supplementary information:

- o Line 104 “duplicating” do you mean multiplying

Response:

This has now been corrected to “multiplying”.

- o Line 105 “weighed” do you mean weighted

Response:

This has now been corrected to “weighted”.

- o Line 128 “weighing” do you mean weighted

Response:

This has now been corrected to “Weighting”.

Reviewer #2 (Remarks to the Author):

The authors have analysed 86 lesions from 53 metastatic melanoma patients to study the genomic evolution of metastatic progression in this cohort. Using high-depth WES (Whole Exome Sequencing) data and copy number profiling, the authors first examined the genomic complexity of the lesions, which they define as the fraction of the genome in an aberrant state, and conclude that samples in which whole genome duplication (WGD) is evident (approx. 40%), are more genomically complex. Additionally, patients whose tumours display evidence of WGD, display a larger copy

number diversity, defined as the mean number of copy number alterations separating samples from individual patients, but overall patients displayed a low degree of mutational diversity. The authors go on to assess mutational processes, showing that trunk mutations are dominated by UV signature, polyclonal seeding and the influence of therapy in these samples, and propose a model for the progression of melanoma metastasis wherein the acquisition of driver mutations, such as BRAF, is usually followed by a gain in the mutated allele, and in which WGD is an event that happens mostly after the UV-induced mutation but before any copy number alteration.

The authors present an in-depth analysis of the genomic landscape of a cohort of melanoma metastases, and their findings are in keeping with previous studies. However, a number of issues would need to be addressed for their conclusions to be adequately supported:

- The authors have concluded, based on data from metastatic samples only, that, as they say in the abstract, metastases may arise from different subclones in the primary tumour. However, they have no evidence of this, given they have not analysed any primary tumour material to confirm this. Their extrapolation of their data to positing theories about tumour evolution is somewhat problematic, as at best, they are only comparing intra-patient metastatic lesions, rather than primary to metastatic evolution

Response:

Regrettably, we did not have primary tumor tissue available for the present study. We concur with the reviewer that our findings provide no definitive evidence of metastases arising from different subclones in the primary tumor. Thus, we have modified the wording when interpreting these data (abstract and section 2, Discussion, last sentence). However, we believe that although other explanations for our observations may be possible, they are not likely. Especially, we do not expect seemingly UV-related DNA-damage to occur after metastatic spread to organs other than the skin. It should also be noted, that analyzing primary tissue would not necessarily provide definitive answers to this hypothesis either; absence of distinct subclones with private mutations in the primary could be due to sampling of the wrong geographical region of the tumor or it could be that the subclone is very small, thereby escaping the detection limit for sequencing.

- Whilst the authors claim to present 86 lesions from 53 patients, these include a number of samples from acral, mucosal, and uveal melanoma, from which only a single lesion was analysed. The bulk of the study pertains to 23 patients from whom multiple lesions were obtained, and whilst the authors present some analysis of the single lesions in figure 1, they have not inferred any conclusions for these samples later in the rest of the study. In fact, the authors have not shown or made any conclusions about whether the evolution progression model proposed by them works for different melanoma types and single lesions. Having other types of melanomas in the cohort doesn't provide any novel insight.

Response:

We believe including single lesions and melanomas from non-sun-exposed areas to add several interesting points to the results. First, although previously observed by others, our finding (Results, section one, end) that all metastatic lesions from primaries of unknown origin revealed a mutational signature suggesting origin

from sun-exposed areas, is of significance. Further, including all tumors provided a number allowing description of driver frequency (Results, section 2), mutation range, as well as mutation signatures (Results, section 3). Single lesion patients are also included in the timing of genomic alterations. Also, our detection of GNAQ and GNA11 as drivers in uveal melanomas confirms findings by others.

We fully concur that it had been of interest to evaluate potential progression models for the rare subtypes separately. Regrettably, however, the number of such tumours in our material do not allow for this type of analyses. Still, for the reasons mentioned above, we believe that the additional information extracted from single lesions, as well as the more rare tumour types, add important information to our study.

- In the method section for copy number profiling, authors have extensively used the simulated data to tune parameters. Apart from the distribution type, the details on the simulated data are lacking. Further, in the estimating allelic tumor copy numbers and tumor purity section, equations do not have any prior algorithms referenced along with them. This makes it hard to comprehend how the equations were derived. Lastly, without the availability of the scripts/codes for copy number pipeline it becomes difficult to assess the advantages of this pipeline.

Response:

As noted in our response to Reviewer#1, we realize that a more detailed description of the performance on our in-house copy number tool should be made available to the reader. This has now been duly dealt with in the Supplementary information. See also further details in our response to Reviewer#1's first point.

Authors have performed mutational signature analysis separately for trunk and branch mutations. The signature analysis for branch mutation is affected by increased residual sum of squares error in model fitting when the numbers of branch mutations are less than 50 (as also shown in Fig S7). Hence the results are affected by mathematical uncertainty in parameter estimation. Further, as shown in the Figure S7 (b), only one patient has predominance of UV signature, hence the conclusion that some branch mutations fit into UV pattern is flawed.

Response:

We concur with this comment, also raised by reviewer 3, that there are issues of uncertainty, related to mutational signature analysis when the number of mutations is limited. Thus, we have modified the text accordingly, pointing to these limitations when presenting the data (Results section "Shift in mutational processes"; line 8-13) as well as in the methods description (Supplementary methods, section "Mutational signature analysis"). However, we find that some of the enriched mutations contributing to the signatures are so typically UV-related (e.g. YC>T) that any other interpretation than UV-damage is less likely. As such, we believe that these findings provide information of relevance and interest, for which reason we would like to keep the information in the manuscript, but now also pointing clearly toward the limitations.

Overall it seems to the reviewer that a key weakness of this study is the lack of evidence from primary tumour material to support their proposed model. Conclusions have been drawn on somewhat circumstantial evidence, providing little advancement in knowledge of metastatic melanoma evolution. The authors acknowledge that previous studies have described genomic alterations in melanoma progression, and claim to advance knowledge about distant metastases, but given there appears little difference between physically proximal and distant metastases in this study, the reviewer does not find the current study provides and advancement on these previous studies.

Response:

We respectfully disagree with the reviewer's assessment here. We make several observations that have not been adequately addressed previously. As for the similarity of molecular features between distant and regional metastases, we believe this similarity to represent a novel finding per se, not to be confused with previous knowledge.

Minor points

At several places, tables from the main paper are missing and all the supplementary tables are not labelled.

Response:

These issues are now corrected.

There are several typos and mismatched references for the figures and tables in the paper. For example in driver mutations and genomic complexity section it should have been Fig 1(c) instead of fig 1(d).

Response:

These issues are now corrected.

Reviewer #3 (Remarks to the Author):

The authors perform a comparative analysis of 86 distant metastatic lesions from 53 melanoma patients (multiple lesions available for 23/53). In order to characterize the genomic alterations underlying disease progression, they perform whole exome sequencing and identify coding mutations and copy number changes. Integrating the two data, they show that mutated BRAF allele is usually amplified before whole-genome duplication (WGD), which happens before metastatic spread. They also conclude what WGD is correlated with greater genomic complexity in comparison with those tumors without WGD. They characterize the subclonal structure and determine the relationship between different tumor sites from the same patient. They identify trunk as well as branch mutations and report that all driver mutations are on the trunk. Comparing the variant allele fraction of private mutations to the mutations

on the trunk, they conclude that there is no evidence of polyclonal seeding in these patients.

Cohorts of this many patients with multiple metastatic samples available are rare and overall the analysis presented is sound and of interest to other researchers in the field of cancer genomics. However, there are some minor points that needs to be addressed:

1. To characterize subclonal structure, the authors first determine if a mutation is shared across all sites (trunk) or private to a particular sample (private). They correct variant allele frequency (VAF) for locus-specific copy number and tumor purity to calculate relative VAF and compare the shared and private mutations to identify private mutations that are clonal. They conclude that existence of private clonal mutations indicates monoclonal seeding. Although the inference of the monoclonality is probably true, the authors should perform a statistical clustering of relative VAF across different samples from the same patient to identify mutation clusters. There are many published methods to do this kind of analysis including PyClone that the authors say they chose not to use. What is the reason why the authors opted out of using such an algorithm?

Response:

This is a fair point raised by the Reviewer. We have previously run a mutation clustering algorithm (PyClone) on our samples as part of our data analyses. Doing so, we found what we consider to be overcalling of subclonal populations in many patients in our dataset (e.g. MM17, MM21 and MM32). This may be seen as “subclones” based on low numbers of mutations, for which the inferred clonal structure in related samples is incompatible, where a mutation found to represent a daughter clone in one sample, represented a parent clone in another. The reason for this may be related to resolution restrictions given that we have analyzed WES-data and not higher resolution data sets such as genome sequencing or SNP arrays, as PyClone should ideally be run using the latter¹. We therefore saw it as necessary to assess subclonality following a strategy more suitable to our specific dataset, being careful in the application of mutation clustering across samples.

Still, as PyClone (or a similar mutation clustering procedure) is a common practice, recognized by researchers in the field, we have now included results from PyClone analyses of multi-sample patients in the study. This is now specified in the main text in the section “Evaluation of polyclonal seeding”, with the results summarized in figure S9, and the results of a particular patient (MM61) presented in figure S10a. We do, however view these results largely as supporting information, adding validity to our findings, which are still mainly based on our initial approach.

2. Figure 2b is difficult to understand. The range of the y-axis for branch and trunk mutations are different. Representation of this information would be best done with phylogenetic trees. Alternatively the author could include a third panel with a bar plot showing the proportion of branch/trunk mutations per sample.

Response:

We realize that our original Figure 2b perhaps was not clear enough, because of the different Y-axes. In order to make this figure more easily accessible to the reader, we have now modified the lower panel to include the total number of

mutations, with the different types (trunk, branch, private) marked by colors. In this way, the lower panel shows the relative abundance of trunk and branch mutations, while the upper panel still illustrates the details of variable number of branch mutations between samples (and patients).

While we also realize that phylogenetic trees are the most common way of depicting such data, we believe figure 2 to adequately represent the quantitative information concerning mutational heterogeneity. Still, phylogenetic trees may add to the visualization (e.g. enabling comparison of branch lengths between samples individual samples), and we therefore also now include phylogenetic trees for all multi sample patients (Supplementary figure S4). As the data thus presented is somewhat redundant with the original Supplementary figure S4 in, we have omitted this previous version of the figure from the revised manuscript.

3. The information in Figure 4a should be plotted for each sample pair in a way similar to Figure S10a. This will clearly show that trunk mutations have relative VAF of 1 in the two samples under comparison and there are indeed clonal private mutations in most samples. If space is an issue, this could be provided as a supplementary figure.

Response:

We agree that such a modified figure may clarify the point to the reader and have now amended figure 4a to include a pairwise comparison of the samples in the figure (similar to the previous Figure S10; now Figure S11).

4. For mutation signature analysis, a minimum of 10 mutations is not enough to perform reliable signature assignment. It might be better to leave this analysis out and only show the information summarized in Figure 3, which is already clearly indicating a change in mutational landscape in later subclones.

Response:

This is an important issue also raised by reviewer 2. Although the generation of signatures from a limited number of observations should be done with great care, such a signature is obvious in 1 patient. Also, taking all limitations into account, we believe the fact that there are indications of such signatures in 5 additional samples is worth mentioning. However, we have now modified the text (Results section “Shift in mutational processes”; line 8-13), clearly pointing to the limitations of the analysis. See also our response to Reviewer#2.

5. Did the authors interrogate the copy number changes for driver alterations?

Response:

Our assessment of copy numbers as mechanisms potentially involved in driving events was restricted to copy number changes in segments encompassing driver mutations. Here, the only significant finding was related to BRAF. In the present study, we did not attempt to identify copy number alterations as driving events per se. This might be an interesting task, but we believe that this dataset (based on WES) is not well suited for such an analysis. One would potentially require WGS with more exact determination of breakpoints to perform such an analysis properly.

6. Could supplementary table 1 be extended to include sample-level information such tumor purity etc.?

Response:

A new table with sample-level information has now been included (Supplementary table S3).

1. Roth A, *et al.* PyClone: statistical inference of clonal population structure in cancer. *Nat Methods* **11**, 396-398 (2014).

Figure Rev01 (next page): Comparison of CNA-calling by ASCAT and the in-house algorithm applied in the present study.

REVIEWERS' COMMENTS:

Reviewer #1 (Remarks to the Author):

I am satisfied that the authors have addressed the major concerns I raised in the review process. This inclusion of further supplementary information will allow the reader an informed view of the analysis processes employed.

Reviewer #2 (Remarks to the Author):

The authors have addressed most of our previous concerns related to novelty of the results and the in-house tool developed for the analysis. They have provided a detailed description of the tool in the supplementary material. The results have been correlated with previous findings. The study provides important information such as the monoclonal origin of the metastatic lesions. They also correlate whole genome duplicated tumours (WGD) with greater genomic complexity in comparison with tumours without the WGD.

Reviewer #4 (Remarks to the Author):

POINT-1: To characterize subclonal structure, the authors first determine if a mutation is shared across all sites (trunk) or private to a particular sample (private). They correct variant allele frequency (VAF) for locus-specific copy number and tumor purity to calculate relative VAF and compare the shared and private mutations to identify private mutations that are clonal. They conclude that existence of private clonal mutations indicates monoclonal seeding. Although the inference of the monoclonality is probably true, the authors should perform a statistical clustering of relative VAF across different samples from the same patient to identify mutation clusters. There are many published methods to do this kind of analysis including PyClone that the authors say they chose not to use. What is the reason why the authors opted out of using such an algorithm?

Response:

This is a fair point raised by the Reviewer. We have previously run a mutation clustering algorithm (PyClone) on our samples as part of our data analyses. Doing so, we found what we consider to be

overcalling of subclonal populations in many patients in our dataset (e.g. MM17, MM21 and MM32). This may be seen as “subclones” based on low numbers of mutations, for which the inferred clonal structure in related samples is incompatible, where a mutation found to represent a daughter clone in one sample, represented a parent clone in another. The reason for this may be related to resolution restrictions given that we have analyzed WES-data and not higher resolution data sets such as genome sequencing or SNP arrays, as PyClone should ideally be run using the latter¹. We therefore saw it as necessary to assess subclonality following a strategy more suitable to our specific dataset, being careful in the application of mutation clustering across samples.

Still, as PyClone (or a similar mutation clustering procedure) is a common practice, recognized by researchers in the field, we have now included results from PyClone analyses of multi-sample patients in the study. This is now specified in the main text in the section “Evaluation of polyclonal seeding”, with the results summarized in figure S9, and the results of a particular patient (MM61) presented in figure S10a. We do, however view these results largely as supporting information, adding validity to our findings, which are still mainly based on our initial approach.

Response to rebuttal:

This point is adequately addressed. As the authors suggest, the reason why methods like PyClone is over-calling subclonal mutation cluster in this dataset is because the data is from WES and there are few mutations in the branches. I agree that PyClone results summarized in figure S9 and S10 provide support for their analysis (monoclonal original of the mets).

POINT-2: Figure 2b is difficult to understand. The range of the y-axis for branch and trunk mutations are different. Representation of this information would be best done with phylogenetic trees. Alternatively the author could include a third panel with a bar plot showing the proportion of branch/trunk mutations per sample.

Response:

We realize that our original Figure 2b perhaps was not clear enough, because of the different Y-axes. In order to make this figure more easily accessible to the reader, we have now modified the lower panel to include the total number of mutations, with the different types (trunk, branch, private) marked by colors. In this way, the lower panel shows the relative abundance of trunk and branch mutations, while the upper panel still illustrates the details of variable number of branch mutations between samples (and patients).

While we also realize that phylogenetic trees are the most common way of depicting such data, we believe figure 2 to adequately represent the quantitative information concerning mutational heterogeneity. Still, phylogenetic trees may add to the visualization (e.g. enabling comparison of branch lengths between samples individual samples), and we therefore also now include phylogenetic trees for all multi sample patients (Supplementary figure S4). As the data thus

presented is somewhat redundant with the original Supplementary figure S4 in, we have omitted this previous version of the figure from the revised manuscript.

Response to rebuttal:

This point is adequately addressed. The new panel in F2 is easier to follow now. However, there seems to be a problem in the caption labeling.

POINT-3: The information in Figure 4a should be plotted for each sample pair in a way similar to Figure S10a. This will clearly show that trunk mutations have relative VAF of 1 in the two samples under comparison and there are indeed clonal private mutations in most samples. If space is an issue, this could be provided as a supplementary figure.

Response:

We agree that such a modified figure may clarify the point to the reader and have now amended figure 4a to include a pairwise comparison of the samples in the figure (similar to the previous Figure S10; now Figure S11).

Response to rebuttal:

This point is adequately addressed.

POINT-4: For mutation signature analysis, a minimum of 10 mutations is not enough to perform reliable signature assignment. It might be better to leave this analysis out and only show the information summarized in Figure 3, which is already clearly indicating a change in mutational landscape in later subclones.

Response:

This is an important issue also raised by reviewer 2. Although the generation of signatures from a limited number of observations should be done with great care, such a signature is obvious in 1 patient. Also, taking all limitations into account, we believe the fact that there are indications of such signatures in 5 additional samples is worth mentioning. However, we have now modified the text (Results section "Shift in mutational processes"; line 8-13), clearly pointing to the limitations of the analysis. See also our response to Reviewer#2.

POINT-5: Did the authors interrogate the copy number changes for driver alterations?

Response:

Our assessment of copy numbers as mechanisms potentially involved in driving events was restricted to copy number changes in segments encompassing driver mutations. Here, the only significant finding was related to BRAF. In the present study, we did not attempt to identify copy number alterations as driving events per se. This might be an interesting task, but we believe that this dataset (based on WES) is not well suited for such an analysis. One would potentially require WGS with more exact determination of breakpoints to perform such an analysis properly.

Response to rebuttal:

This point is adequately addressed.

POINT-6: Could supplementary table 1 be extended to include sample-level information such tumor purity etc.?

Response:

A new table with sample-level information has now been included (Supplementary table S3).

Response to rebuttal:

This point is adequately addressed.